# Trapped ion mobility spectrometry and PASEF enable in-depth lipidomics from minimal sample amounts

Catherine G. Vasilopoulou [1], Karolina Sulek[2], Andreas-David Brunner [1], Ningombam Sanjib Meitei[3], Ulrike Schweiger-Hufnagel[4], Sven W. Meyer[4], Aiko Barsch[4], Matthias Mann [1,2]* & Florian Meier [1]*

A comprehensive characterization of the lipidome from limited starting material remains very challenging. Here we report a high-sensitivity lipidomics workflow based on nanoflow liquid chromatography and trapped ion mobility spectrometry (TIMS). Taking advantage of parallel accumulation–serial fragmentation (PASEF), we fragment on average 15 precursors in each of 100 ms TIMS scans, while maintaining the full mobility resolution of co-eluting isomers. The acquisition speed of over 100 Hz allows us to obtain MS/MS spectra of the vast majority of isotope patterns. Analyzing 1 µL of human plasma, PASEF increases the number of identified lipids more than three times over standard TIMS-MS/MS, achieving attomole sensitivity. Building on high intra- and inter-laboratory precision and accuracy of TIMS collisional cross sections (CCS), we compile 1856 lipid CCS values from plasma, liver and cancer cells. Our study establishes PASEF in lipid analysis and paves the way for sensitive, ion mobility-enhanced lipidomics in four dimensions.

---

[1] Max Planck Institute of Biochemistry, Martinsried, Germany. [2] NNF Center for Protein Research, Copenhagen, Denmark. [3] PREMIER Biosoft, Indore, India. [4] Bruker Daltonik GmbH, Bremen, Germany. *email: mmann@biochem.mpg.de; fmeier@biochem.mpg.de

Disentangling the lipid composition of biological model systems and clinical samples in a robust and high throughput manner promises novel insight into basic biology, as well as the onset and progression of disease[1–4]. Lipid extracts from biological sources can be analyzed either directly via high-resolution mass spectrometry[5,6] or via online liquid chromatography coupled to mass spectrometry (LC-MS)[7]. Lipid identifications base on accurate mass and the MS[2] or MS[3] fragmentation pattern, which is increasingly facilitated by recent software developments and ever growing reference databases[8–11]. Established LC-MS lipidomics workflows separate lipids at flow rates in the higher micro- or milliliter per minute range, which ensures high sample throughput and robustness, but also compromises sensitivity. As the available sample amount becomes a limiting factor, for example with small tissue biopsies from biobanks or small cell sub-populations, it is increasingly attractive to employ nanoflow chromatography[12–14].

MS technology has greatly improved and state-of-the-art high-resolution Orbitrap or time-of-flight (TOF) instruments transmit ions very efficiently and achieve low- to sub-ppm mass accuracy[15,16]. The high acquisition speed of TOF analyzers makes them compatible with very fast separation techniques such as ion mobility spectrometry (IMS)[17,18]. Nested in-between LC and MS, IMS provides an additional dimension of separation based on the ions' shape and size (collisional cross section, CCS). This is particularly interesting for lipidomics, as it provides an opportunity to separate otherwise unresolved isomers[19–22]. Furthermore, the chemical structure of lipids is closely linked to the CCS, which allows predictions by machine learning and could facilitate lipid identification[23–26].

Trapped ion mobility spectrometry (TIMS) is a relatively new form of IMS that inverts the separation principle of classical drift tube ion mobility[27–31]. Ions entering the TIMS analyzer are positioned in an electrical field by the drag of a gas flow. Lowering the electrical force releases ions from the TIMS device separated by their ion mobility, while the IMS resolution is proportional to the user-defined ramp time. It can be tuned to over 200 CCS/ΔCCS, for example to separate isomeric lipids with distinct double bond positions or geometries[32]. We recently introduced a MS scan mode termed parallel accumulation serial fragmentation (PASEF) that synchronizes TIMS with MS/MS precursor selection[33]. In proteomics, PASEF increases MS/MS scan rates more than tenfold, importantly, without the loss of sensitivity that is otherwise inherent to faster acquisition rates[34].

Here, we explore whether the PASEF principle can be transferred to lipidomics. We build on nanoflow chromatography to establish a rapid PASEF lipidomics workflow capable of comprehensively analyzing low sample amounts. To investigate the potential of the additional TIMS dimension, we set out to compile a high-precision lipid CCS library from body fluids, tissue samples, and human cell lines.

## Results

**Development of the nanoflow PASEF lipidomics workflow**. We aimed to develop a rapid workflow that enables global lipid analysis in a straightforward manner (Fig. 1). We adapted an MTBE lipid extraction protocol[35] that is applicable to common biological sample types, such as body fluids, tissue, as well as cell lines (Fig. 1a) and requires only a few manual liquid handling steps that could easily be automated in the future. We found that our extraction protocol scales well from small sample volumes (1 µL blood plasma) to relatively large cell counts (5e5 HeLa cells) and can be performed in <1 h.

We loaded the lipid extracts directly onto a $C_{18}$ column and eluted them within 30 min, for a total of little more than 1 h

analysis time per sample when using both positive and negative ionization modes (Fig. 1b). Retention times were reproducible with median CVs of 0.54% in replicate injections prior to alignment (Supplementary Data 1) and chromatographic peak widths were in the range of 3–6 s full width at half maximum (FWHM), at least two orders of magnitude slower than TIMS ion mobility analysis (100 ms) and the acquisition speed of high-resolution TOF mass spectra (~100 µs). The timsTOF Pro mass spectrometer (Bruker Daltonics) features a dual TIMS analyzer that allows to utilize up to 100% of the incoming ion beam[29] (Methods). In this mode, ions are accumulated in the first TIMS analyzer while another batch of ions is separated by ion mobility in the second TIMS analyzer. TIMS closely resembles classical drift tube ion mobility, however, ions arrive at the mass analyzer in the inverse order, which means low-mobility (and high-mass) ions are released first, followed by ions with higher mobility (and lower mass). In our experiments with 100 ms TIMS scan time, the ion current accumulated during 100 ms was concentrated into ion mobility peaks of 2–3 ms FWHM, which should lead to a 50-fold increase in signal-to-noise as compared with continuous acquisition. These peak widths equate an ion mobility resolution of 40–50 CCS/ΔCCS. The ion mobility-resolved mass spectra can be illustrated in two-dimensional heat maps, from which suitable precursor ions are selected for fragmentation in data-dependent MS/MS mode (Fig. 1b). With PASEF, multiple precursors are fragmented in each TIMS ramp by rapidly switching the quadrupole (see below). As the collision cell is positioned after the TIMS analyzer in the ion path, fragment ions occur at the same ion mobility position as their precursor ions in MS1 mode.

We make use of this information in the post-processing (Fig. 1c) to connect MS/MS spectra to their corresponding MS features extracted from the four-dimensional data space (retention time, *m/z*, ion mobility, intensity). Finally, we rely on established lipidomics software to automatically assign lipids to the spectra based on diagnostic fragment ions and database matching (Methods). Our workflow automatically converts ion mobility to CCS values and records them for all detected MS1 features and thus all identified lipids.

**Evaluating PASEF in lipidomics**. The central element of our workflow is the PASEF acquisition method. PASEF takes advantage of the temporal separation of ions eluting from the TIMS device to select multiple precursors for MS/MS acquisition in a single TIMS scan[33]. To illustrate, Fig. 2a, b shows representative MS1 heat maps of co-eluting lipids from an LC-TIMS-MS analysis of human plasma. The ions are widely spread in *m/z* vs. ion mobility space, while higher mass roughly correlates with lower ion mobility. In standard MS/MS mode, the quadrupole isolates a single precursor mass throughout the entire TIMS scan time (red dots in Fig. 2a). However, the targeted precursor completely elutes during about 6 out of 100 ms, and thus over 90% of the acquisition time is effectively not used. In PASEF mode, the quadrupole instead switches its mass position within ~1 ms to capture as many precursors as possible (red dots in Fig. 2b). In this example, 16 precursors were selected during a single PASEF scan, which translates into a 16-fold increased MS/MS acquisition rate of over 100 Hz. Importantly, this does not come at a loss in sensitivity because the full precursor ion signal of the 100 ms accumulation time is captured.

In a 30 min analysis of plasma, we found that on average 15 precursors were fragmented per PASEF scan (Fig. 2c), confirming that the PASEF principle is transferable to lipidomics. In total, we acquired 187,177 MS/MS spectra—15-fold more than without PASEF. This fragmentation capacity greatly exceeds the number of expected lipids and, in principle, allows to acquire MS/MS

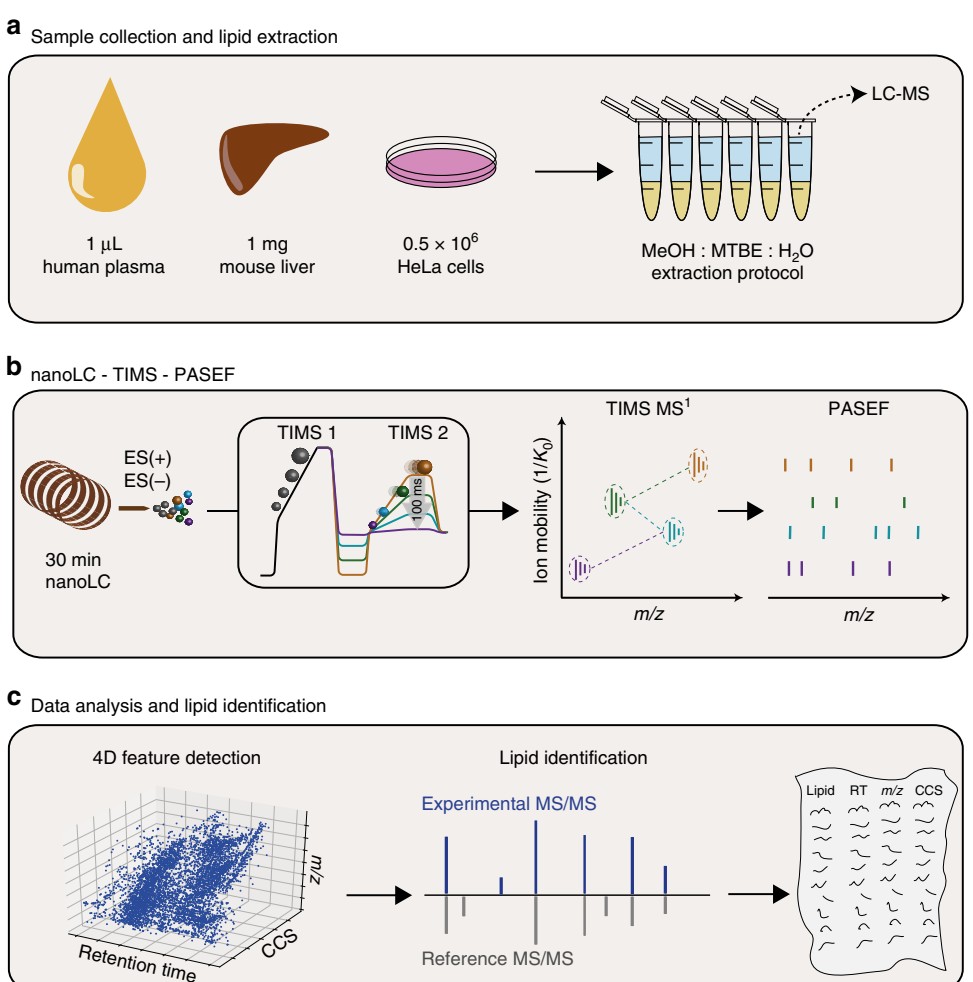

**Fig. 1 Nanoflow lipidomics with trapped ion mobility spectrometry. a** Lipids from various biological sources, such as body fluids, tissues and cells, are analyzed using a single MeOH:MTBE extraction. **b** The crude extract is injected into a nanoflow liquid chromatography (LC) system coupled online to a high-resolution TIMS quadrupole time-of-flight mass spectrometer (timsTOF Pro). In the dual TIMS analyzer, ions are accumulated in the front part (TIMS 1), while another batch is released as a function of ion mobility from the TIMS 2 analyzer. PASEF synchronizes precursor selection and ion mobility separation, which allows fragmenting multiple precursors in a single TIMS scan at full sensitivity. **c** Features are extracted from the four-dimensional (retention time, $m/z$, ion mobility, intensity) data space and assigned to PASEF MS/MS spectra for automated lipid identification and compilation of comprehensive lipid CCS libraries. MeOH = methanol, MTBE = methyl-tert-butyl ether, CCS = collisional cross section.

spectra for every suitable isotope pattern detected in a single lipidomics LC run. Here, we chose to fragment low-abundance precursors repeatedly to increase their signal-to-noise ratios in a summed spectrum. On average, precursors were fragmented two times as indicated by the acquisition engine.

We evaluated the performance of our PASEF method with lipid extracts from human plasma, mouse liver, and HeLa cells (Fig. 2d and Supplementary Fig. 1). In all three sample types, the 4-dimensional feature detection yielded 8900–13,400 MS1 features above the intensity threshold and after collapsing multiple adducts. In standard TIMS-MS/MS mode, on average 5.5% of these were fragmented. This fraction increased up to 11.5-fold with PASEF and, in both negative and positive ionization mode, about 65% of all features had corresponding MS/MS spectra. Overall, PASEF increased the number of identified lipids across all runs on average 3.6-fold (Supplementary Fig. 2). To further investigate whether PASEF is fast enough to acquire MS/MS spectra of close to all informative lipid features in a short time, we extended the LC gradients to 60 and 90 min (Supplementary Fig. 3). Indeed, 97% of all lipids identified with the three times longer gradient were already identified in the 30 min PASEF run, which confirms our hypothesis and suggests that even shorter runtimes could be explored.

**Comprehensive and accurate lipid quantification.** Having ascertained that PASEF achieves a very high MS/MS coverage of lipidomics samples, we investigated our automated data analysis pipeline in more detail (Fig. 3a). Starting from the thousands of 4D features detected in all replicate injections of human plasma, mouse liver and HeLa cells, we kept those with assigned MS/MS scans for further analysis. PASEF spectra are resolved by ion mobility and the software extracts the MS/MS spectra at the ion mobility position of the respective precursor ion. We then searched all MS/MS spectra considering four lipid categories with the respective lipid classes and subclasses. This yielded 653–1595 annotations for each sample and ionization mode. We manually inspected all automatically annotated MS/MS spectra to filter potential false positives based on the observed fragmentation pattern (Methods). Finally, we grouped adducts, isomers, and co-eluting peaks that were separated by their ion mobility but could not be distinguished based on their MS/MS spectra. However, we

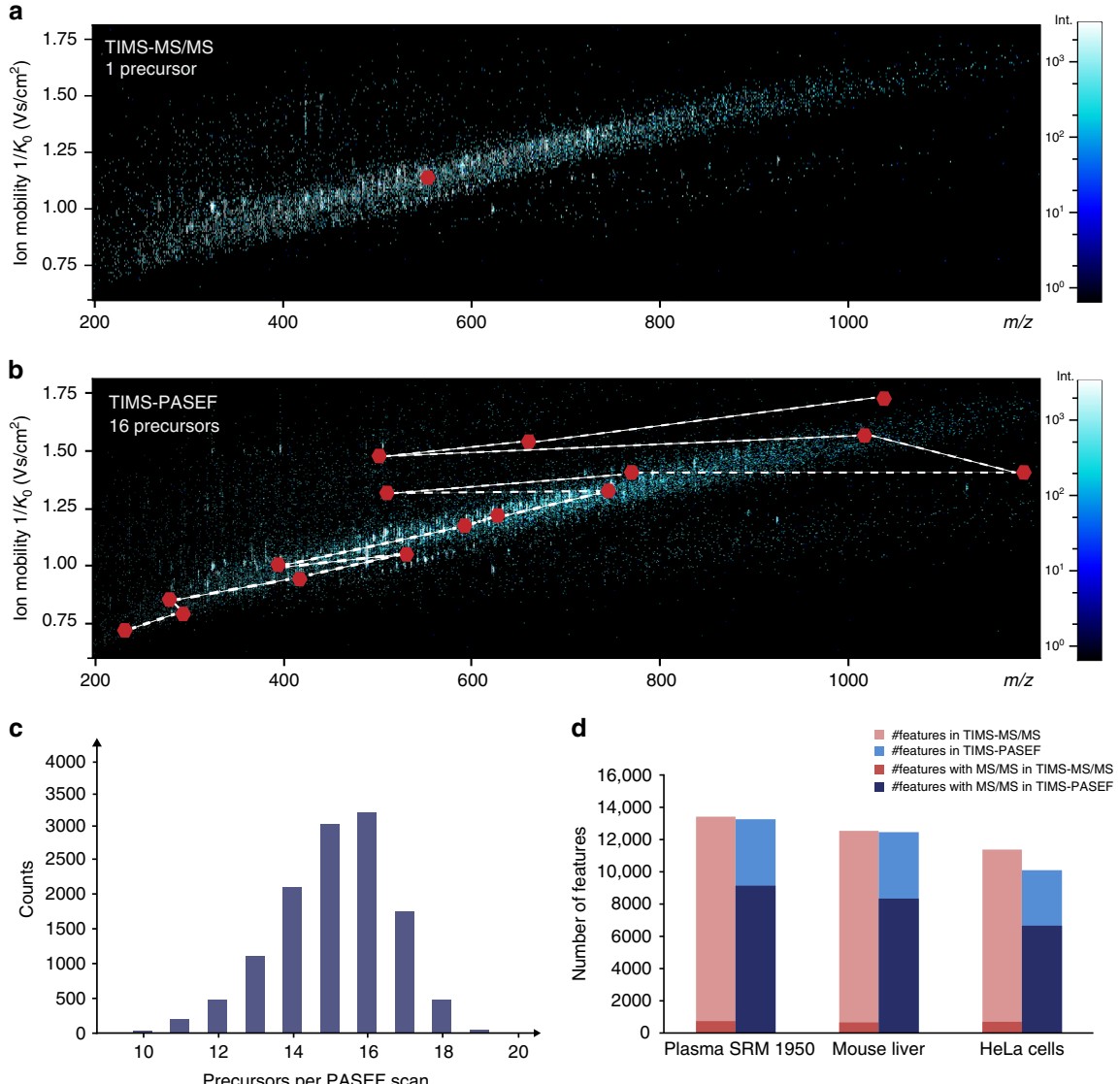

**Fig. 2 Evaluating PASEF in lipidomics.** Heat-map visualization of a representative trapped ion mobility resolved mass spectrum of human plasma at an elution time of 9.2 min. Red dots indicate precursors selected for MS/MS fragmentation in the subsequent 100 ms PASEF scan in **a**, standard TIMS-MS/MS mode and **b** TIMS-PASEF mode. The dashed line indicates the positioning of the quadrupole. **c** Distribution of the number of precursors per PASEF scan in an LC-MS analysis of human plasma lipid extract ($n = 1$). **d** Total number of 4D features extracted from 30 min runs of human plasma ($n = 5$), mouse liver ($n = 5$), and human cancer cells ($n = 5$) in positive ion mode without (TIMS-MS/MS, red) and with PASEF (TIMS-PASEF, blue). The fraction of features with assigned MS/MS spectra is indicated by a darker color.

kept separate lipids with same MS/MS-based annotation but eluting in close proximity as these are potential isomers. Removing duplicates resulted in 460–879 identified unique lipids per experiment. Combining both ionization modes, we identified 1108 unique lipids from the equivalent of 0.05 μL plasma per injection, 976 unique lipids from 10 μg liver tissue and 1351 unique lipids from ~2000 HeLa cells, with a median absolute precursor mass accuracy of 1.06 ppm (Supplementary Data 2–4). The identified lipid species covered all major lipid classes such as glycerophospholipids (PC, PE, PA, PS, PI, PG), oxidized glycerophospholipids, monoacyl-, diacyl- and triacyl-glycerols, sterol lipids, ceramides, glycosphingolipids, and phosphosphingolipids.

This comprehensive lipid coverage from relatively small sample amounts motivated us to investigate our sensitivity limit in more detail. Starting from the concentration above, we diluted the lipids extracted from human SRM 1950 plasma over three orders of magnitude in seven steps. With a 10-fold dilution, we were still

able to identify 526 lipids in positive mode and this number dropped below 400 only at >100-fold dilution (Supplementary Fig. 4). We reasoned that this sensitivity is partially due to our nanoflow chromatography setup as opposed to conventional high-flow systems. In fact, a direct comparison indicated a 100-fold lower sensitivity limit with nanoflow in both ionization modes. Injecting the same amounts of plasma lipids on column, we identified three to six times more lipid species with the nanoflow setup (Supplementary Fig. 4).

In biological or clinical studies, quantitative accuracy is at least as important as cataloging the lipid composition and may be compromised if lipids are sparsely detected across samples. We hypothesized that the speed of PASEF and the improved signal-to-noise of TIMS should lead to very reproducible quantification. Indeed, we observed that 816 out of the total 976 identified lipids in liver tissue were quantified in five out of five replicates (Fig. 3b), resulting in a data completeness of 95.4%. The median

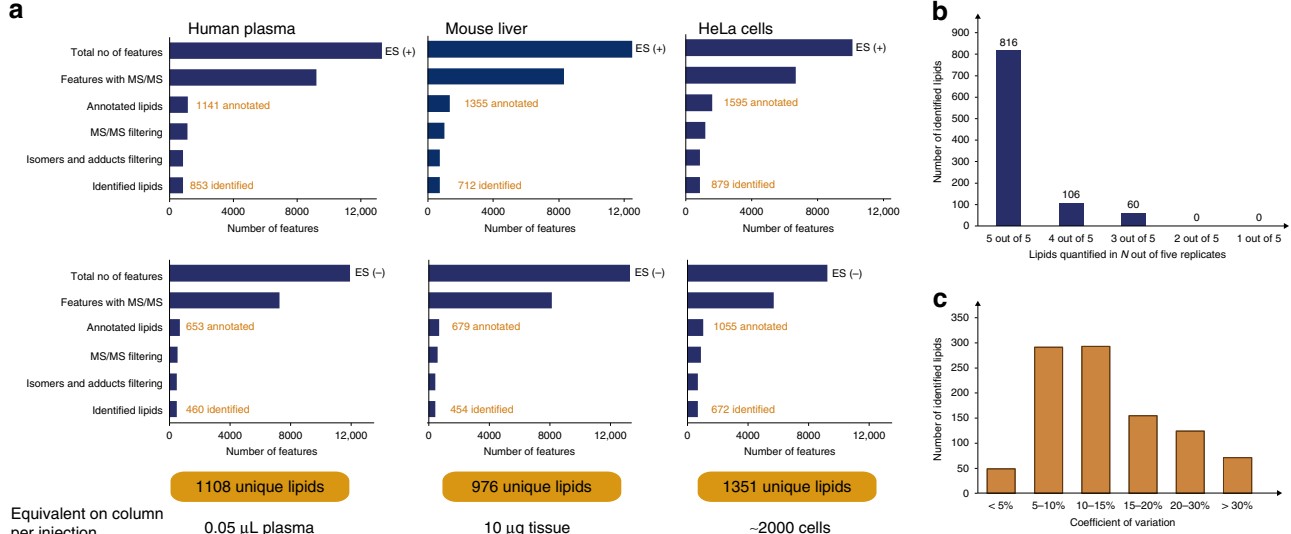

**Fig. 3 Lipid identification and quantification. a** Sequential data analysis steps from the total number of detected 4D features to unique lipids for human plasma, mouse liver, and human cancer cells in both ionization modes. **b** Number of lipids quantified in *N* out of five replicate injections of liver tissue extract. **c** Coefficients of variation for 976 lipids quantified in at least three out of five replicate injections of liver tissue extract.

coefficient of variation (CV) was 12.3%, and 80% of all quantified lipids had a CV below 20% (Fig. 3c and Supplementary Data 5).

From these results, we conclude that our nanoflow PASEF lipidomics workflow covers the lipidome comprehensively with high quantitative accuracy, while requiring only minimal sample amounts.

**Coverage of the human plasma lipidome.** To assess the lipid coverage of our PASEF workflow in more detail, we analyzed the human plasma Standard Reference Material (SRM 1950), a pool from 100 individuals from the United States in the age range of 40–50 years, provided by the National Institute of Diabetes and Digestive and Kidney Diseases and the National Institute of Standards (NIST)[36]. This sample has served as a reference for many lipidomics studies, establishing a range of detectable lipid species and their absolute concentrations[37]. The LIPID MAPS consortium recently compiled consensus results from 31 laboratories, each of which followed their in-house analysis workflow (Bowden et al.[38]). In an effort to disentangle the human plasma lipidome, Quehenberger et al.[39] employed class-specific analysis strategies to quantify about 500 lipids from >1 mL NIST SRM 1950 plasma.

Taking these two studies as a reference, we first compared the number of identified lipids in each lipid category based on the short name annotation (Fig. 4a and Supplementary Data 6, 7). Starting from 1 µL plasma and with a single extraction protocol, our PASEF workflow detected many more glycerolipids and glycerophospholipids, exceeding both studies three- to four-fold. At the same time, 87 and 83% of all glycerolipids and glycerophospholipids reported in the Bowden et al. study[38] were also present in our dataset. Similarly, we retrieved 65 and 49% of all lipids from these two abundant plasma lipid categories reported by Quehenberger et al.[39]. We observed a two-fold gain for sphingomyelins, again with a high overlap of 77 and 60% with both reference studies. Analysis of ceramides typically requires specific extraction methods and this category was therefore underrepresented in ours as well as in the Bowden et al. study[38] relative to the class-specific analysis by Quehenberger et al.[39]. From another analytically challenging class of lipids, sterol lipids, we still detected 33 species in the human plasma reference sample.

To further investigate the sensitivity of our method, we mapped all identified lipids onto absolute plasma concentrations reported in ref. [38] (Fig. 4b and Supplementary Data 7). We quantified about 80% of the lipids covering the full abundance range from about 0.01 up to 1000 nmol/mL. For example, we achieved full coverage of the triacylglycerols and also quantified less abundant lipids such as phosphatidylethanolamines comprehensively. Even though coverage was sparser in the lowest abundance range, we quantified the least abundant lysophosphatidylcholine (LPC 22:1) with a reference concentration of 0.013 nmol/mL. Since we injected only 1/20th of the lipids extracted from 1 µL plasma in each replicate, this translates into a sensitivity in the attomol range for the entire workflow.

**Accuracy and precision of online lipid ^TIMSCCS measurements.** In addition to generating MS/MS spectra for almost all detectable precursors with PASEF, TIMS measures the ion mobility of all identified and unidentified lipids. We calibrated all TIMS measurements to reduced ion mobility values using well-characterized phosphazine derivatives and converted them to collisional cross sections (^TIMSCCS) using the Mason–Schamp equation[29,40]. Because they have the same underlying physics, ^TIMSCCS can be directly related to drift tube experiments[30,31,41] and should result in high quality and highly reproducible collisional cross section data.

First, we investigated a mixture of commercially available lipid standards (Differential Ion Mobility System Suitability Lipidomix Kit, Avanti) on four timsTOF Pro instruments in two independent laboratories (in Bremen and Munich, Germany) (Fig. 5a). The measured ^TIMSCCS values for all 22 lipid ions (12 distinct lipids) clustered closely around their median values with a median CV of 0.35%. The median intra-instrument variability in five replicate injections ranged from 0.10 to 0.17% and the median intra-laboratory CV was between 0.18 and 0.21% in both laboratories. The ^TIMSCCS values were also highly reproducible between laboratories with an average inter-laboratory CV of only 0.35% and did not reveal any lipid-class specific biases (Supplementary Data 8).

To test whether the high quality of ^TIMSCCS values manifests in complex biological samples, we first investigated all detectable features in all three sample types regardless of their identification

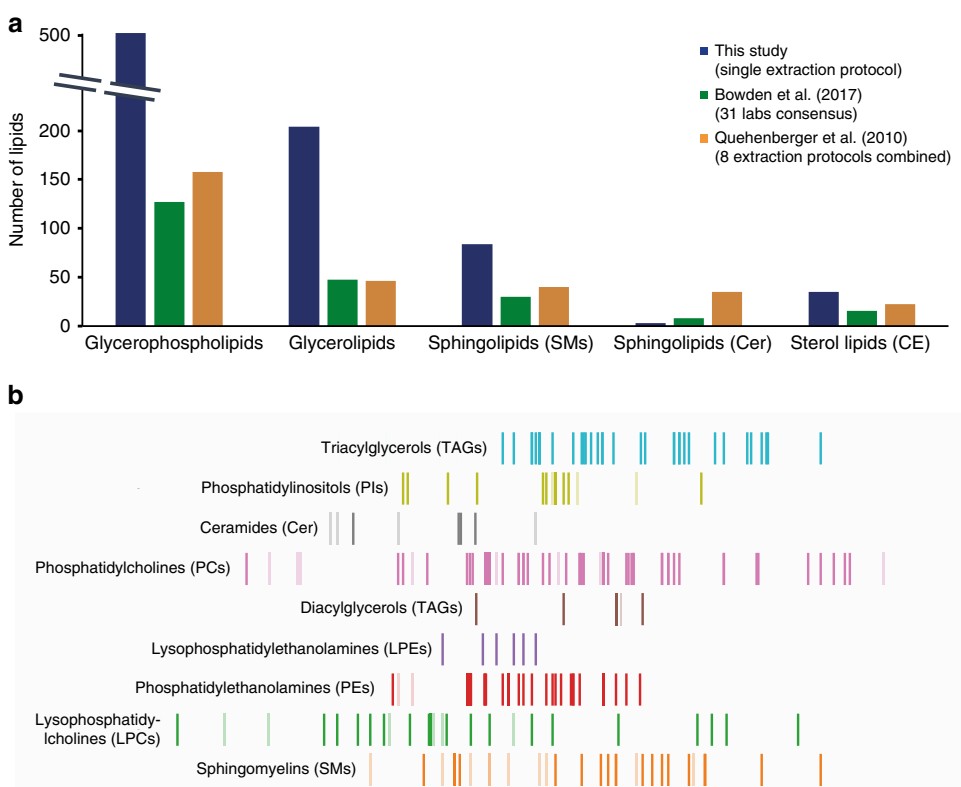

**Fig. 4 Analysis of 1 μL NIST SRM 1950 human plasma. a** Number of identified lipids from major lipid classes in this study and two reference studies from the same standard material[38,39]. **b** Mapping of lipids identified with our PASEF lipidomics workflow to absolute plasma concentrations reported in [38]. Vertical lines indicate the abundance range of reported lipids from different lipid classes and dark color indicates commonly identified lipids in both studies.

as a lipid. Plotting the $^{TIMS}$CCS values across repeated injections on one instrument revealed excellent reproducibility (Pearson $r >$ 0.99) (Supplementary Fig. 5 and Supplementary Data 9), which motivated us to measure lipid extracts from the NIST SRM 1950 plasma on all four instruments. Considering only $^{TIMS}$CCS of lipids identified in all experiments, we found median CVs < 0.1 to 0.45% in repeated injections on the same instrument (Fig. 5b and Supplementary Data 10) and similar intra-laboratory CVs of 0.15–0.45%. Overall, CCS measurements from both laboratories agreed within 0.38% on average and were highly correlated with a Pearson correlation coefficient $r >$ 0.99 for all pair-wise comparisons (Fig. 5c and Supplementary Data 11).

Having ascertained highly reproducible $^{TIMS}$CCS measurements in complex samples, we next investigated the accuracy of our results by comparing it with different methods and instrumentation. Our dataset shared 149 and 28 lipid identifications (based on the short name annotation) with recent reports from the Zhu[24] and McLean[23] laboratories, which both employed drift tube ion mobility analyzers to establish high-precision reference data. Our comparison revealed a very high correlation (Pearson $r =$ 0.999) and 98% of all values were within ±1% deviation centered at zero (Fig. 5d and Supplementary Data 12). The median absolute deviation was 0.18%, which is very well in the range of recently reported inter-laboratory variability for standard compounds measured with a commercial drift tube analyzer[42].

The high reproducibility of lipid ion mobility measurements makes them also very attractive for machine learning approaches.

The Zhu laboratory has developed a support vector regression model that predicts lipid CCS values from SMILES structures[24] and which is implemented in the Bruker MetaboScape software in a modified version (Methods). Even though the model was trained on independent data from a different instrument type, predicted and experimental $^{TIMS}$CCS values correlated very well (Pearson $r =$ 0.996) and the relative deviations were normal distributed with 95% of the values within ±2% deviation for lipid classes that were contained in the initial training data (Fig. 5e and Supplementary Data 13). The median absolute deviation was 0.44% and based on the experimental precision demonstrated above, we expect that machine learning models trained directly on TIMS data yield even more accurate predictions.

**The TIMS lipidomics landscape.** Data generated by our TIMS lipidomics workflow span a three-dimensional data space in which each feature is defined by retention time, $m/z$ and CCS, with intensity as a 4th data dimension. To explore this data space, we compiled all measurements from human plasma, mouse liver and HeLa cells acquired in both ionization modes. The total dataset comprises CCS values of 1856 unique lipids (positive mode), representing the four major lipid categories and 15 lipid classes (Supplementary Data 14). To make our dataset fully accessible, we provide Supplementary Data 14 in a format that follows the standard lipid nomenclature guidelines by the LIPID MAPS consortium[43] and the Lipidomics Standards Initiative (https://lipidomics-standards-initiative.org/).

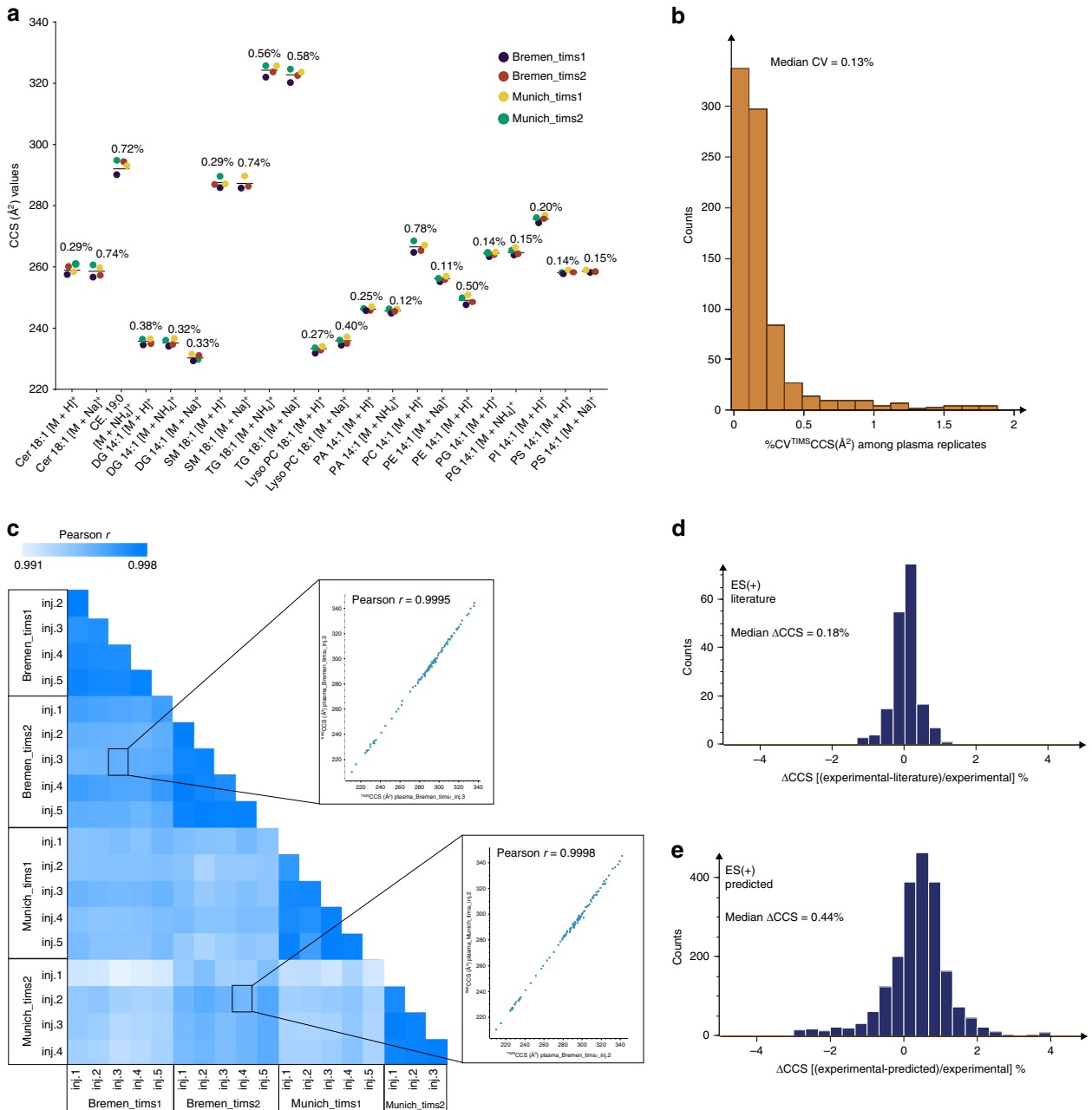

**Fig. 5 Precise and accurate determination of lipid $^{TIMS}$CCS values. a** Cross-instrument and cross-laboratory $^{TIMS}$CCS measurement of a mixture of standard compounds (source data provided in Supplementary Data 8). Data labels indicate the coefficient of variation (CV) ($n = 4$ instruments). **b** CVs of $^{TIMS}$CCS values for lipids commonly identified in replicate injections of a human plasma sample ($n = 5$ replicates, $n = 1$ instrument). **c** Pair-wise correlation of lipid $^{TIMS}$CCS values from human plasma SRM 1950 measured on four different timsTOF Pro instruments. Relative deviation of experimental $^{TIMS}$CCS values in this study (**d**) from literature reports[23,24] and **e** machine learning predictions[24].

The investigation of the correlation of lipid mass and ion mobility has been a long term interest in ion mobility spectrometry-based lipidomics[23,24]. TIMS and PASEF provide a very efficient way to extend the scope of such studies to complex biological samples. Figure 6a shows a three-dimensional representation of all identified lipids in all three sample types in positive ionization mode color-coded by their respective classes. Each lipid class occupies a discrete space in the conformational landscape, which reflects the structural differences in their chemical composition. Hydrophilic lipids, such as monoacyl- and low molecular weight diacylglycerophospholipids ($m/z$ 400–600) that elute first in reversed-phase chromatography, distribute in the CCS dimension from 204 to 253 Å$^2$. The second

half of the LC gradient is dominated by the large population of glycerolipids and glycerophospholipids, which are often co-eluting, but distinct in mass and CCS. For example, diacylglycer-ols (DAG) and triacylglycerols (TAG) differ by one acyl chain and occupy a different CCS space shifted by 54.7 Å$^2$. Similarly, the head groups can strongly influence the ion mobility of lipids, as exemplified by PIs and PGs with the same acyl chain composition (Fig. 6a and Supplementary Fig. 6).

A key feature of our workflow is that each detected MS feature is precisely positioned in the multi-dimensional data cuboid, while the speed of PASEF ensures that most of these features are associated with MS/MS information. We hypothesized that the combined information can be leveraged to assign putative lipid

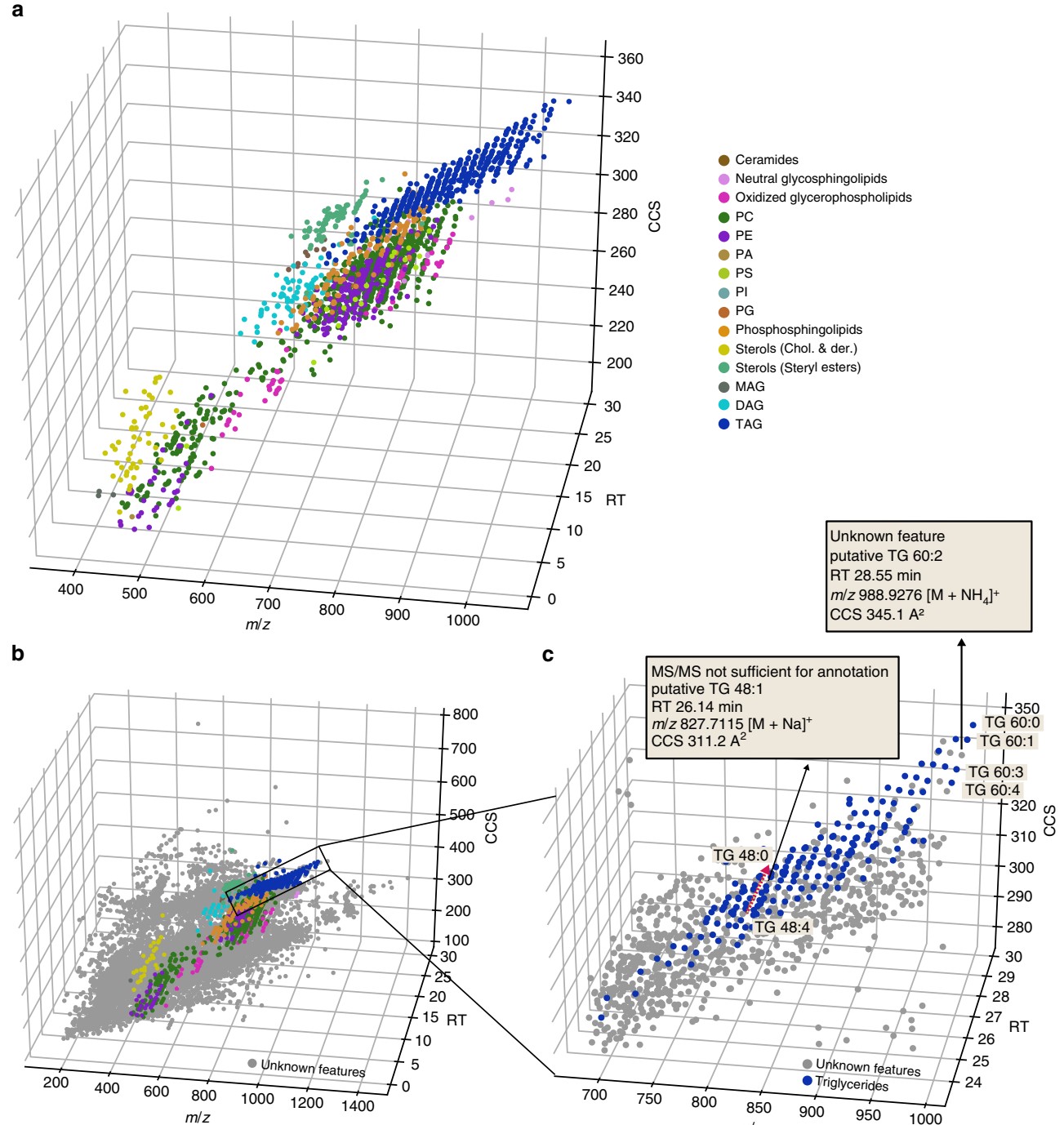

**Fig. 6 The conformational landscape of lipid ions in TIMS. a** Three-dimensional (RT, *m/z*, CCS) distribution of 1856 lipids from various classes from three biological samples (plasma, liver, HeLa) in positive ion mode. **b** Overlay of unidentified (gray) and identified MS features detected in a human plasma sample. **c** Zoom into the data cuboid and putative assignment of two previously unidentified lipids based on their relative position in the data space. PC = Phospatidylcholine, PE = Phospatidylethanolamine, PA = Phosphatidic acid, PI = Phospatidylinositol, PG = Phospatidylglycerol, PS = Phospatidylserine, MAG = Monoacylglycerol, DAG = Diacylglycerol, TAG = Triacylglycerol.

identifications for features that would otherwise have remained unidentified. To test this, we overlaid all detected features with MS/MS information on top of all identified lipids (Fig. 6b). Zooming into the distinct space occupied by triglycerides revealed the conformational fine-structure of this lipid class, which results in clusters of lipids with the same acyl chain composition (Fig. 6c). Within each cluster, the lipids are differentiated by their degree of unsaturation as the addition of a double bond decreases the CCS value almost linearly. This enabled the identification of features that were not fully characterized by the available MS/MS information. As an example, we putatively assigned the MS feature at retention time 26 min, *m/z* 827.7115 ($\Delta = 1.9$ ppm) and CCS 311.2 Å$^2$ as TG 48:1 based on the relative position in the 3D space. This is further supported by the predicted CCS value of 308.4 Å$^2$, which deviates <1% from our experimental value. Similarly, we derived a putative assignment for TG 60:2, which

had escaped identification due to a low-quality MS/MS spectrum in this particular experiment.

## Discussion

Trapped ion mobility spectrometry is a particularly compact and efficient ion mobility setup, in which ions are held against an incoming gas flow and released as a function of their size and shape. The Bruker timsTOF Pro incorporates two TIMS analyzers in the front part of a high-resolution quadrupole TOF mass spectrometer and fully supports our recently introduced PASEF scan mode. In this study, we developed a nanoflow lipidomics workflow that takes full advantage of TIMS and PASEF.

In 100 ms TIMS scans, readily compatible with fast chromatography, we made use of the ion mobility separation to fragment on average 15 precursors per PASEF scan by rapidly switching the mass position of the quadrupole. Even though this translates into MS/MS acquisition rates above 100 Hz, the ion count per spectrum, and thus the sensitivity, is determined by the TIMS accumulation time (here 100 ms or 10 Hz). In principle, this allows to acquire MS/MS spectra for all detectable isotope patterns in short LC-MS runs, which would otherwise require much longer gradients or multiple injections and advanced acquisition strategies. While many of the acquired spectra remained unidentified in our current data analysis pipeline, the PASEF acquisition strategy generates very comprehensive digital MS/MS archives of all samples that can be mined with alternative and novel search algorithms at any time in the future.

Our results indicate that the nanoflow PASEF lipidomics workflow is readily applicable to a broad range of biological samples, such as body fluids, tissue samples and cell cultures. With a single extraction step, we quantified thousands of lipids with very high accuracy and reproducibility from as little as micrograms of tissue or only a few thousand cells. This makes our workflow very attractive for sample-limited applications, such as lipid analysis from biopsies, micro dissected tissue or sorted cell populations.

An important application of lipidomics is the investigation of body fluids, for example blood plasma. Our analysis of a human reference sample was in good agreement with previous reports and greatly surpassed the coverage of glycerol- and glycerophospholipids, using only a fraction of the analysis time and sample amount. Based on these results, we estimate a limit of detection in the attomole range for these analyte classes.

Our online PASEF lipidomics workflow positions each lipid in a four-dimensional space with a precision of 1 ppm for masses, <0.2% for CCS and about 1% for retention times. We exemplified that this precision and accuracy can be leveraged to facilitate lipid assignment in addition to the comprehensive MS/MS information generated by PASEF. To further explore this data space, we compiled a library of over 1800 high-precision lipid CCS values directly from unfractionated biological samples. Our dataset largely extends the number of reported lipid CCS values and provides a basis for emerging machine learning techniques to predict CCS values more accurately and for a broader range of lipid classes.

We conclude that TIMS and PASEF enable highly sensitive and accurate lipidomics, and generate comprehensive digital archives of all detectable species along with very precise ion mobility measurements—a wealth of information which awaits full exploration and application. We also note that all the analytical advantages demonstrated for lipidomics should carry over to metabolomics in general, an area that we are currently exploring.

## Methods

**Chemicals and biological samples**. 1-Butanol (BuOH), iso-propanol (IPA), ortho-phosphoric acid, formic acid, methanol (MeOH), and water were purchased from Fisher Scientific (Germany), and methyl tert-butyl ether (MTBE) from Sigma Aldrich (Germany) in analytical grade or higher purity. The standard lipid mixture Differential Ion Mobility System Suitability Lipidomix Kit was purchased from Avanti Polar Lipids, Inc (product no. 330708). The human plasma reference standard NIST SRM 1950 was obtained from Sigma Aldrich. Human cancer cells (HeLa, human, ACC57, DSMZ) were cultured in Dulbecco's modified Eagle's medium (DMEM), with 10% fetal bovine serum, 20 mM glutamine and 1% penicillin–streptomycin (all from PAA Laboratories, Germany) and collected by centrifugation. The cell pellets were washed, frozen in liquid nitrogen and stored at −80 °C. Mouse liver was dissected from an individual male mouse (strain: C57BL/6) and snap frozen immediately. Animal experiments were performed in compliance with the ethical and institutional regulations of the Max Planck Institute of Biochemistry for animal testing and research, and have been approved by the government agencies of Upper Bavaria.

**Lipid extraction**. Lipids were extracted using an adapted MTBE protocol[35]. Plasma samples were thawed on ice and the sample preparation was performed at 4 °C. 200 μL cold MeOH were added to 1 μL of blood plasma and vortexed for 1 min. Subsequently, 800 μL of cold MTBE were added and the sample was mixed for another 6 min before adding 200 μL water. To separate the organic and aqueous phases, we centrifuged the mixture at 10,000 g for 10 min at 4 °C. The upper organic phase was collected and vacuum-centrifuged to dryness. To extract lipids from mouse liver, we first homogenized 1 mg of tissue in methanol and followed the extraction protocol described above. To extract lipids from ~5 × 10^5 HeLa cells, they were lysed after addition of MTBE by sonication (Bioruptor, Diagenode, Belgium). The dried lipid extracts from all samples were reconstituted in BuOH:IPA:water in a 8:23:69 ratio (v/v/v) with 5 mM phosphoric acid (nanoflow LC)[14] or in MeOH: Dichloromethane 9:1 (v/v) (high-flow LC) for LC-MS analysis.

**Liquid chromatography**. An Easy-nLC 1200 (Thermo Fisher Scientific) ultra-high pressure nanoflow chromatography system was used to separate lipids on an in-house reversed-phase column (20 cm × 75 μm i.d.) with a pulled emitter tip, packed with 1.9 μm $C_{18}$ material (Dr. Maisch, Ammerbuch-Entringen, Germany). The column compartment was heated to 60 °C and lipids were separated with a binary gradient at a constant flow rate of 400 nL/min. Mobile phases A and B were ACN:$H_2O$ 60:40% (v/v) and IPA:ACN 90:10% (v/v), both buffered with 0.1% formic acid and 10 mM ammonium formate. The 30 min LC-MS experiment started by ramping the mobile phase B from 1 to 30% within 3 min, then to 51% within 4 min and then every 5 min to 61, 71 and 99%, where it was kept for 5 min and finally decreased to 1% within 1 min and held constant for 2 min to re-equilibrate the column. The total LC runtime was ~40 min including time for re-filling of LC pumps and sample loading before the start of the analytical gradient.

The gradient was extended proportionally for 60 min and 90 min experiments. We injected 1 μL in positive and 2 μL in negative ion mode on column and each sample was injected five times in both ionization modes.

For the high-flow experiments (flow rate 0.4 mL/min), an Elute-HT UHPLC system (Bruker Daltonics Bremen, Germany) was used with an Intensity $C_{18}$ column (100 mm × 2.1 mm, 1.9 μm beads) (Bruker, Daltonics, Germany) heated to 55 °C. Mobile phases and gradient were the same as with the nano-flow setup. The injection volume was 5 μL and each sample was injected five times in both ionization modes.

**Trapped ion mobility—PASEF mass spectrometry**. The nanoLC was coupled to a hybrid trapped ion mobility-quadrupole time-of-flight mass spectrometer (timsTOF Pro, Bruker Daltonics, Bremen, Germany) via a modified[16] nano-electrospray ion source (Captive Spray, Bruker Daltonics). For a detailed description of the mass spectrometer, see Refs. [33,34]. Briefly, electrosprayed ions enter the first vacuum stage where they are deflected by 90° and accumulated in the front part of a dual TIMS analyzer. The TIMS tunnel consists of stacked electrodes printed on circuit boards with an inner diameter of 8 mm and a total length of 100 mm, to which an RF potential of 300 V_{PP} is applied to radially confine the ion cloud. After the initial accumulation step, ions are transferred to the second part of the TIMS analyzer for ion mobility analysis. In both parts, the RF voltage is superimposed by an electrical field gradient (EFG), such that ions in the tunnel are dragged by the incoming gas flow from the source and retained by the EFG at the same time. Ramping down the electrical field releases ions from the TIMS analyzer in order of their ion mobility for QTOF mass analysis. The dual TIMS setup allows operating the system at 100% duty cycle, when accumulation and ramp times are kept equal[29]. Here, we set the accumulation and ramp time to 100 ms each and recorded mass spectra in the range from m/z 50–1550 in both positive and negative electrospray modes. The ion mobility was scanned from 0.6 to 1.95 Vs/cm². Precursors for data-dependent acquisition were isolated within ± 1 Th and fragmented with an ion mobility-dependent collision energy, which was linearly increased from 25 to 45 eV in positive mode, and from 35 to 55 eV in negative mode. The overall acquisition cycle of 0.4 s comprised one full TIMS-MS scan and three PASEF MS/MS scans. Low-abundance precursor ions with an intensity above a threshold of 100 counts but below a target value of 4000 counts were repeatedly scheduled and otherwise dynamically excluded for 0.2 min. TIMS ion charge control was set to 5e6. The TIMS dimension was calibrated linearly using four selected ions from the

Agilent ESI LC/MS tuning mix [$m/z$, $1/K_0$: (322.0481, 0.7318 Vs cm$^{-2}$), (622.0289, 0.9848 Vs cm$^{-2}$), (922.0097, 1.1895 Vs cm$^{-2}$), (1221,9906, 1.3820 Vs cm$^{-2}$)] in positive mode and [$m/z$, $1/K_0$: (666.01879, 1.0371 Vs cm$^{-2}$), (965.9996, 1.2255 Vs cm$^{-2}$), (1265.9809, 1.3785 Vs cm$^{-2}$)] in negative mode.

**Dilution series experiment**. Plasma SRM 1950 was extracted on the same day and injected in both nano- and high-flow LC systems as described above. For positive mode experiments with the high-flow system, 30 µL plasma was extracted and reconstituted in 300 µL reconstitution solvent to inject a final volume of 5 µL (translating into 0.5 µL plasma on column). For the nanoflow separation, 1 µL plasma was extracted and reconstituted in 20 µL reconstitution solvent to inject a final volume of 1 µL (translating into 0.05 µL plasma on the column). These stock samples were then sequentially diluted in 1:3.3, 1:10, 1:33, 1:100, 1:333, and 1:1000 ratios (vol:vol). For the high-flow experiment in negative mode, 30 µL plasma was extracted and reconstituted in 30 µL to inject a final volume of 5 µL (5 µL plasma on column). For the nanoflow experiment, 1 µL plasma was extracted and reconstituted in 20 µL to inject a final volume of 2 µL (0.1 µL plasma on column). The samples were diluted to the final concentrations as above. All samples were injected in five replicates.

**Data analysis and bioinformatics**. The mass spectrometry raw files were analyzed with MetaboScape alpha version 5.0 (Bruker Daltonics, Germany). This version contains a novel feature finding algorithm (T-ReX 4D) that automatically extracts data from the four-dimensional space ($m/z$, retention time, ion mobility and intensity) and assigns MS/MS spectra to them. Masses were recalibrated with the lock masses $m/z$ 622.028960 (positive mode) and $m/z$ 666.019887 (negative mode) and the ion mobility dimension was recalibrated using the ions of the tuning mix as above. Feature detection was performed using an intensity threshold of 500 counts in positive mode and 200 counts in the negative mode. The minimum number of data points in the 4D TIMS space was set to 100, or 50 when using recursive feature extraction.

Lipid annotation of detected molecular features with assigned MS/MS spectra was performed using the high-throughput lipid search (HTP) function of SimLipid v6.05 software (PREMIER Biosoft, Palo Alto, USA). The lipid search comprised four lipid categories, Glycerolipids (GL), Glycerophospholipids (GP), Sphingolipids (SP) and Sterol lipids (SL) and TAG, DAG, PA, PC, PE, PG, PI, PS, Ceramides, Sphingomyelins, Neutral Glycosphingolipids, Steryl esters, Cholesterols, and Derivatives, as well as oxidized glycerophospholipid classes. PE and PC lipids with ether- and plasmalogen- substituents were considered. Lipid species from TAG and sterol classes were not considered for the negative mode MS/MS database search. Glycerophospholipids were only considered if containing an even number of carbons on at least one of the fatty acid chains. We searched for [M + H]$^+$, [M + Na]$^+$, and [M + NH$_4$]$^+$ ions in positive mode, and [M–H]$^-$, [M + Cl]$^-$, [M–CH$_3$]$^-$, [M + HCOO]$^-$ and [M + AcO]$^-$ in negative mode. The precursor ion and MS/MS fragment mass tolerances were set to 5 and 10 ppm, respectively.

The initial search results were filtered to ensure that lipids were annotated based on high-quality MS/MS spectra with fragment ions corresponding to structure specific characteristic ions. To this end, we manually inspected the SimLipid results and removed potential false positives and refined lipid annotations based on head-groups and/or fatty acyl composition as follows. In positive mode, we rejected unlikely PC lipid ion species such as [PC + NH$_4$]$^+$ and [PC + Na]$^+$ if the corresponding [M + H]$^+$ ion was not observed, and if [M-59 + Na]$^+$ (neutral loss of (CH$_3$)$_3$N) and [M-183 + Na]$^+$ (neutral loss of phosphocholine) fragments were not detected. Lipids from GP, ST, and SP categories were required to have their corresponding head group diagnostic ions e.g., $m/z$ 369.3516 for cholesterol esters, $m/z$ 184.073 for PC lipid species, as well as the neutral loss of 141 Th for PEs. Neutral glycosphingolipids and ceramides were rejected if the structure-specific N ″-type fragments were not annotated. However, lipid species from sterol classes were accepted if the precursor ion was the base peak in the MS/MS spectrum. TG/DG lipids with three or two unique fatty acid chains were reported only if at least two/one fatty acid chain fragment ion were/was detected. In negative mode, we rejected all lipid annotations for which we did not detect at least one characteristic fragment ion corresponding to one of the fatty acid chains.

We report lipid identifications with increasing level of fragment ion evidence using the following nomenclature: (i) a short name (e.g., PC 32:1) to indicate mass-resolved lipid molecular species, (ii) a long name for composition-resolved identifications where the symbol @ indicates that this particular acyl-chain is not fully characterized by fragment ions (e.g., Cer d18:1_26:0@), and (iii) a long name where head group and fatty acyl-chain composition are fully characterized (e.g., PG 16_1:16:1). Note that sn1/sn2/sn3 chain assignments, positions of the double bonds, as well as cis/trans isomers are not evident from our data and therefore not annotated.

Lipid CCS values were predicted in MetaboScape using SMILES from LipidMaps based on a support vector machine learning approach by Zhou et al.[24]. Mass spectrometric metadata such as the PASEF frame MS/MS information were extracted from the.tdf files using an SQLite database viewer (SQLite Manager v3.10.1). Further data analysis and visualization was performed in Python 3 (Jupyter Notebook) and Perseus (v1.6.0.8)[44].

**Reporting summary**. Further information on research design is available in the Nature Research Reporting Summary linked to this article.

## Data availability
The mass spectrometry raw files have been uploaded to the MASS Spectrometry Interactive Virtual Environment (MassIVE) and are accessible via the identifier MSV000083858. Source data for all figures are provided in the Supplementary Data 1–14. All other data are available from the corresponding authors on reasonable request.

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

## Acknowledgements

We thank our colleagues in the Department of Proteomics and Signal Transduction for discussion and help, in particular D. Voytik, A. Strasser, J. Mueller, I. Paron, and C. Deiml. We acknowledge our colleagues H. Neuweger, N. Kessler, and S. Wehner at Bruker Daltonics in Bremen for their support in data processing and analysis. This work was partially supported by the German Research Foundation (DFG–Gottfried Wilhelm Leibniz Prize), the Novo Nordisk Foundation (Grant agreement NNF14CC0001), the European Commission (FP7 GA ERC-2012-SyG_318987–ToPAG), and the Max-Planck Society for the Advancement of Science.

## Author contributions

C.G.V., K.S., M.M., and F.M. designed the research project; C.G.V., K.S., A.D.B., S.M., A.B., and F.M. performed the experiments; C.G.V., K.S., N.S.M., U.S.H., S.M., A.B., and F.M. analyzed the data; N.S.M. as well as U.S.H., S.M., and A.B. contributed analytical tools; C.G.V., M.M., and F.M. wrote the paper.

## Competing interests

The following authors state that they have potential conflicts of interest regarding this work: U.S.H., S.M., and A.B. are employees of Bruker, the manufacturer of the timsTOF Pro, and N.S.M. is employee of PREMIER Biosoft, the vendor of the SimLipid software. The other authors declare no competing interests.
