## [Peer Review File · Nature Communications]

Reviewers' comments:

Reviewer #1 (Remarks to the Author):

The manuscript by Vasilopoulou et al describe the use of trapped ion mobility mass spectrometry (TIMS) with parallel accumulation-serial fragmentation (PASEF) for lipidomics studies of plasma, cells, and liver tissue. The advantages of TIMS and PASEF have both been recently described in the literature and this manuscript describes the depth of coverage and new information that can be obtained in lipidomics through their combination. Generally speaking, the manuscript is well written – although there are several typographical errors that this reviewer is confident will be remedied through editing (e.g. Page 17 “...trained direly on...”).

The prevailing sentiment of this reviewer centers on if these studies are a little premature for publication in Nature Communications. In particular, the use of TIMS as it relates to the reporting of collision cross sections. While there is some recent evidence that TIMS cross sections can be related to drift tube cross sections further validation studies should be provided. The reported intra-laboratory CV values appear to be quite good, however data shown in Supplementary Figure 4 may demonstrate a fair amount of error associated with individual species across two replicate measurements. These replicate errors should be provided in the supplementary table to gauge the accuracy values reported.

The studies here would be reinforced dramatically by validation experiments in both some inter-laboratory studies and through the use of standards similar to those performed using drift tube experiments. This would be highly beneficial to the community as the TIMS technique is quite new and not typical. Few studies exist to lend confidence that the cross sections measured do indeed provide the quality metrics reported by the authors across multiple laboratories. This would provide support that of what other laboratories should expect through adoption of these procedures.

Reviewer #2 (Remarks to the Author):

In this work, the authors utilize nanoLC, and synchronization of trapped ion mobility spectrometry and MS/MS with PASEF, to evaluate the coverage of lipids in small sample sizes of plasma and tissue. While this manuscript was interesting, I was left searching for the novel aspect of this work in order for it to fit into Nature Communications. Many groups have used LC, IMS and MS/MS analyses to

evaluate lipids in complex samples. Also, the difference in lipid groups and double bonds in the LC and IMS separations have been documented in numerous papers. Most groups utilize high flow LC systems for more robust elution times, so the nanoflow aspect was different in this manuscript, but the authors never elaborated on how the nanoflow helped the sensitivity of their analyses and if the elution peak reproducibility was much worse than the high flow systems. The authors also never called out any novel isomers that were detected using the IMS or even what the resolving power of TIMS separations they were using was. While the PASEF seemed to provide more lipid identification when compared to two other manuscripts, they were not significantly greater for all lipid categories (only higher for 2), however they were doing this at lower injection volumes. Also, PASEF has already been shown for proteomics, so this paper just illustrated its utility for another omic system. Since I am having trouble finding the novelty of this paper, I feel other scientist will also have a similar problem. While the results are exciting, as the manuscript is currently written I do not it to have the novelty and broad applicability that someone reading Nature Communication desires. My detailed revisions and comments are listed below.

Revisions and Comments:

1. Introduction, first 2 sentences, pg 3: The first two sentences of the introduction are very basic and should focus on identifying lipids in small sample sizes if that is the focus of this manuscript. The first sentence noting every cell contains lipids should be removed since most people know that lipids make up the cell membrane and provide signaling and storage capabilities. Also, I am not sure why the first two sentences are a separate paragraph.

2. Introduction, pg 4: The authors note the maximum possible resolving power of the TIMS if time is not a constraint. However, what was the resolving power for the system they used under their analysis conditions.

3. Results, pg 5: The authors indicated that “our lipid extraction protocol is applicable to common biological sample types.” What is different about their protocol than those used by most people doing lipidomic analyses.

4. Results, pg 5: When the authors noted a gradient time of 30 min for their LC system and the ability to do positive and negative analyses of each sample in 1 hour, are they not using a wash step?

5. Results, pg 5: What are the CVs for the nanoLC elution time peaks? How do these compare to the higher flow rate systems most people use.

6. Results, pg 6: Why did the authors not use the lipid CCS values from the Zhu lab in their workflow for more possible lipid identifications?

7. Figure 2 caption: What elution time is shown for Figure 2a and 2b?

8. Results, pg 8: In the 111,122 MS/MS spectra, how many of the peaks were analyzed multiple times? How can I use this number to evaluate the method?

9. Results, pg 8: What did acquiring each PASEF scan twice using two different collision energies accomplish? Couldn't the authors use different collision energies based on the drift time location of the ions like the energy ramping other IMS-MS/MS systems utilize?

10. Results, pg 11: What confidence score method was utilized for the lipid identifications? How did the authors know when to "manually" remove a false positive?

11. Results, pg 13: I was confused by the comparison to the other studies. When the authors said they identified 76.2% and 71.2% of all glycerolipids and glycerophospholipids reported in both studies, how was that possible unless each study identified completely different lipids. From Figure 4, it looks like this study identified ~280 glycerophospholipids while Bowden identified ~125 and Quehenberger identified ~160. Even if they had no overlap in lipid identification that is only ~285 total lipids for comparison. Clarification of this whole paragraph is needed as I cannot believe there was no overlapping lipids in the two studies.

12. Results, pg 14: If the authors inject twice as much sample into their instrument, how many lipids are they able to identify? Thus, what is the loss for having such a small injection?

Point-by-point answers to the reviewer comments on “Trapped ion mobility spectrometry (TIMS) and parallel accumulation - serial fragmentation (PASEF) enable in-depth lipidomics from minimal sample amounts”

We thank the editor and reviewers for their careful evaluation of our work. We feel that the constructive comments and additional experiments have improved our manuscript significantly and helped us to further demonstrate the novelty of our approach. In the course of the revision, we have refined our method and improved the overall performance of the entire workflow, which now covers over 1,000 lipids in human plasma, mouse tissue and cancer cell lines from minimal sample amounts and in short time. We measured TIMS collisional cross sections of standard compounds as well as complex lipid extracts on four different instruments, providing new evidence for their very high intra- and inter-laboratory reproducibility. We further include a comparison of high-flow and nano-flow chromatography, highlighting the very high sensitivity of our method. This experiment underlines that the PASEF advantage also applies to widely used setups in other laboratories and enables, for the first time, close to complete MS/MS coverage of all detectable species. As a result, with our novel TIMS-PASEF lipidomics workflow, retention time, precursor mass, collisional cross section and fragment ion spectrum can all be leveraged for lipid identification. We exemplify the general benefits of such comprehensive data for cases that would have remained unidentified in methods lacking one of these dimensions. We have carefully addressed all comments raised by the reviewers as detailed below point-by-point and we hope that the reviewers and editor now find the find our work suitable for publication in *Nature Communications*.

Reviewer #1:

The manuscript by Vasilopoulou et al describe the use of trapped ion mobility mass spectrometry (TIMS) with parallel accumulation-serial fragmentation (PASEF) for lipidomics studies of plasma, cells, and liver tissue. The advantages of TIMS and PASEF have both been recently described in the literature and this manuscript describes the depth of coverage and new information that can be obtained in lipidomics through their combination. Generally speaking, the manuscript is well written – although there are several typographical errors that this reviewer is confident will be remedied through editing (e.g. Page 17 “...trained direly on...”).

We thank the reviewer for the in-depth study of our manuscript and his or her supportive view. We have carefully edited the entire manuscript text to eliminate typographical errors.

The prevailing sentiment of this reviewer centers on if these studies are a little premature for

publication in Nature Communications. In particular, the use of TIMS as it relates to the reporting of collision cross sections. While there is some recent evidence that TIMS cross sections can be related to drift tube cross sections further validation studies should be provided. The reported intra-laboratory CV values appear to be quite good, however data shown in Supplementary Figure 4 may demonstrate a fair amount of error associated with individual species across two replicate measurements. These replicate errors should be provided in the supplementary table to gauge the accuracy values reported.

We agree that TIMS remains a novel technology and we thank the reviewer for challenging us to investigate the collisional cross section measurements in more detail.

Following the reviewer's suggestion, we added a column 'CV (%)' to Supplementary Table S9 associated with Supplementary Fig. 5 (Suppl. Fig. 4 in the first submission), allowing a detailed inspection on the level of individual MS features. This table contains MS features regardless of their identification as lipid. The median CVs were 0.22 to 0.40% and 95% of all values were within an astounding 0.3% in quintuplicate measurement on one instrument in this large scale experiment.

While we do not have access to a drift tube instrument ourselves, we compared our data to publicly available resource data from expert scientists with drift tube ion mobility instruments (revised manuscript p17 and Fig. 5d). The relative deviations are centered on zero and < 1%, which is very well in the 2% range recently reported for inter-lab deviations of drift tube measurements themselves (ref. 43, Stow *et al.*, Anal. Chem. 2017). Furthermore, we did not observe any lipid class-specific bias. To facilitate the inspection of individual lipids, we provide a user-friendly Excel sheet in Suppl. Table S12.

We conclude that TIMS generates high-quality CCS values comparable to conventional drift tube instruments, with the distinct PASEF advantage of very high sensitivity and MS/MS acquisition speed due to parallel ion accumulation and serial fragmentation. We further believe that the compact and robust design of the ~10 cm TIMS device is generally beneficial for the reproducibility of CCS measurements. Regarding the maturity of the technology, we also note that it is generally commercially available, which is more than usually expected of a novel technological development.

Please see also directly below for additional intra- and inter-laboratory comparisons of ^{TIMS}CCS values.

The studies here would be reinforced dramatically by validation experiments in both some inter-laboratory studies and through the use of standards similar to those performed using drift tube experiments. This would be highly beneficial to the community as the TIMS technique is quite new and not typical. Few studies exist to lend confidence that the cross sections measured do indeed provide the quality metrics reported by the authors across multiple laboratories. This would provide support that of what other laboratories should expect through adoption of these procedures.

We highly appreciate the reviewer's suggestion, which we feel adds substantial value to our revised manuscript. In response, we have measured ^{TIMS}CCS values of a commercially available standard lipid mixture as well as lipid extracts from the human SRM plasma on four timsTOF Pro instruments at two different sites (Mann laboratory in Munich and Bruker R&D laboratory in Bremen). Both laboratories used the same TIMS-MS method as detailed in the methods section, while the Bremen laboratory used a high-flow LC and the Mann laboratory used the nanoflow setup. We report our new results on page 16f and in Figure 5 of the revised manuscript:

“First, we investigated a mixture of commercially available lipid standards (Differential Ion Mobility System Suitability Lipidomix Kit, Avanti) on four timsTOF Pro instruments in two independent laboratories (in Bremen and Munich, Germany) (Fig. 5a). The measured ^{TIMS}CCS values for all 22 lipid ions (12 distinct lipids) clustered closely around their median values with a median CV of 0.35%. The median intra-instrument variability in five replicate injections ranged from 0.10 to 0.17% and the median intra-laboratory CV was between 0.18% to 0.21% in both laboratories. The ^{TIMS}CCS values were also highly reproducible between laboratories with an average inter-laboratory CV of only 0.35% and did not reveal any lipid-class specific biases (Suppl. Table S8).

To test whether the high quality of ^{TIMS}CCS values manifests in complex biological samples, we first investigated all detectable features in all three sample types regardless of their identification as a lipid. Plotting the ^{TIMS}CCS values across repeated injections on one instrument revealed excellent reproducibility (Pearson $r > 0.99$) (Suppl. Fig.5 and Suppl. Table S9), which motivated us to measure lipid extracts from the NIST SRM 1950 plasma on all four instruments. Considering only ^{TIMS}CCS of lipids identified in all experiments, we found median CVs <0.1% to 0.45% in repeated injections on the same instrument (Fig. 5b and Suppl. Table S10) and similar intra-laboratory CVs of 0.15-0.45%. Overall, CCS measurements from both laboratories agreed within 0.38% on average and were highly correlated with a Pearson correlation coefficient $r > 0.99$ for all pair-wise comparisons (Fig. 5c and Suppl. Table S11).”

Figure 5. Precise and accurate determination of lipid $TIMS$ CCS values. *a*, Cross-instrument and cross-laboratory $TIMS$ CCS measurement of a mixture of standard compounds. *b*, Coefficients of variation (CV) of $TIMS$ CCS values for lipids commonly identified in replicate injections of a human plasma sample ($n=5$ replicates, $n=1$ instrument). *c*, Pair-wise correlation of lipid $TIMS$ CCS values from human plasma SRM 1950 measured on four different timsTOF Pro instruments. *d,e*, Relative deviation of experimental $TIMS$ CCS values in this study *d*, from literature reports^{24,42} and *e*, machine learning predictions²⁴.

These results further validate the TIMS technology and demonstrate that our results are readily transferable to other laboratories. To further facilitate replication, we deposited executable instrument methods along with all raw MS data files to a public repository. In addition, we summarize all source data conveniently in Supplementary Tables S8 to S11.

Reviewer #2:

In this work, the authors utilize nanoLC, and synchronization of trapped ion mobility spectrometry and MS/MS with PASEF, to evaluate the coverage of lipids in small sample sizes of plasma and tissue. While this manuscript was interesting, I was left searching for the novel aspect of this work in order for it to fit into Nature Communications. Many groups have used LC, IMS and MS/MS analyses to evaluate lipids in complex samples. Also, the difference in lipid groups and double bonds in the LC and IMS separations have been documented in numerous papers. Most groups utilize high flow LC systems for more robust elution times, so the nanoflow aspect was different in this manuscript, but the authors never elaborated on how the nanoflow helped the sensitivity of their analyses and if the elution peak reproducibility was much worse than the high flow systems. The authors also never called out any novel isomers that were detected using the IMS or even what the resolving power of TIMS separations they were using was. While the PASEF seemed to provide more lipid identification when compared to two other manuscripts, they were not significantly greater for all lipid categories (only higher for 2), however they were doing this at lower injection volumes. Also, PASEF has already been shown for proteomics, so this paper just illustrated its utility for another omic system. Since I am having trouble finding the novelty of this paper, I feel other scientist will also have a similar problem. While the results are exciting, as the manuscript is currently written I do not it to have the novelty and broad applicability that someone reading Nature Communication desires. My detailed revisions and comments are listed below.

We are thankful to the reviewer for the interest in our work and the thorough evaluation of our manuscript. However, we profoundly disagree with the assessment of little novelty. The capability to fragment virtually all detected peaks and routinely reduce sample amounts by hundred-fold will have far reaching consequences in the further. We already see this in our clinical work, where we can now measure lipidomics profile for anything that was supplied for proteomics only. The routine availability of extremely reproducible CCS values will likewise have a tremendous impact in the long run.

We have addressed all comments in full detail below, which helped us to highlight the novel aspects and potential impact of our work more clearly in the revised manuscript. In particular, we benchmarked our nanoflow setup against a conventional high-flow system and now demonstrate the benefits of virtually complete PASEF data for lipid identification.

Revisions and Comments:

1. Introduction, first 2 sentences, pg 3: The first two sentences of the introduction are very basic and should focus on identifying lipids in small sample sizes if that is the focus of this manuscript. The first sentence noting every cell contains lipids should be removed since most people know that lipids make up the cell membrane and provide signaling and storage capabilities. Also, I am not sure why the first two sentences are a separate paragraph.

Thank you. We have re-phrased the first two sentences accordingly:

“Disentangling the lipid composition¹ of biological model systems and clinical samples² in a robust manner and with high throughput promises novel insight into basic biology, as well as the onset and progression of disease^{3,4}.”

2. Introduction, pg 4: The authors note the maximum possible resolving power of the TIMS if time is not a constraint. However, what was the resolving power for the system they used under their analysis conditions.

This is indeed an important information that we omitted in our initial manuscript. We now state the experimental ion mobility resolution along with our TIMS settings in the results section (p5):

“In our experiments with 100 ms TIMS scan time, the ion current accumulated during 100 ms was concentrated into ion mobility peaks of 2-3 ms FWHM, which should lead to a 50-fold increase in signal-to-noise as compared to continuous acquisition. These peak widths equate an ion mobility resolution of 40 to 50 CCS/ Δ CCS.”

3. Results, pg 5: The authors indicated that “our lipid extraction protocol is applicable to common biological sample types.” What is different about their protocol than those used by most people doing lipidomic analyses.

Thank you. We have clarified this statement in our revised manuscript:

“We adapted an MTBE lipid extraction protocol³⁵ that is applicable to common biological sample types, such as body fluids, tissue, as well as cell lines (Fig. 1a) and requires only a few manual liquid handling steps that could easily be automated in the future.”

4. Results, pg 5: When the authors noted a gradient time of 30 min for their LC system and the ability to do positive and negative analyses of each sample in 1 hour, are they not using a wash step?

We thank the reviewer for highlighting this point. Indeed, the LC-MS runtime was 30 min in these cases, including a washing step at the end of the gradient. We revised the main text to make this clear and added details in the methods section:

“We loaded the lipid extracts directly onto a C₁₈ column and eluted them within 30 min, for a total of little more than one hour analysis time per sample when using both positive and negative ionization modes (Fig. 1b).”

“The 30 min LC-MS experiment started by ramping the mobile phase B from 1% to 30% within 3 min, then to 51% within 4 min and then every 5 min to 61%, 71% and 99%, where it was kept for 5 min and finally decreased to 1% within 1 min and held constant for 2 min to re-equilibrate the column. The total LC runtime was approximately 40 min including time for re-filling of LC pumps and sample loading before the start of the analytical gradient.”

5. Results, pg 5: What are the CVs for the nanoLC elution time peaks? How do these compare to the higher flow rate systems most people use.

Thank you. We added the CVs for the nanoLC elution time peaks to our revised Supplementary Table 1 and report them in the results section of our revised manuscript. We conclude that the reproducibility of retention times with our nanoflow LC is similar to conventional high-flow setups:

“Retention times were reproducible with median CVs of 0.54% in replicate injections prior to alignment (Suppl. Table S1) and chromatographic peak widths were in the range of 3 to 6 s full width at half maximum (FWHM), ...”

We further took the opportunity to compare high-flow and nano-flow LC in more detail and performed additional experiments with a conventional high flow setup. We included this data in a new paragraph in the results section (p13) and added the new Supplementary Figure 4.

“This comprehensive lipid coverage from relatively small sample amounts motivated us to investigate our sensitivity limit in more detail. Starting from the concentration above, we diluted the lipids extracted from human SRM 1950 plasma over three orders of magnitude in 7 steps. With a 10-fold dilution, we were still able to identify 526 lipids in positive mode and this number dropped below 400 only at greater than 100-fold dilution, the number of identified lipids dropped below 400 (Suppl. Fig. 4). We reasoned that this sensitivity is partially due to our nanoflow chromatography setup as opposed to conventional high-flow systems. In fact, a direct comparison indicated a 100-fold lower sensitivity limit with nanoflow in both ionization modes. Injecting the same amounts of plasma lipids on column, we identified three to six times more lipid species with the nanoflow setup. (Suppl. Fig.4).”

6. Results, pg 6: Why did the authors not use the lipid CCS values from the Zhu lab in their workflow for more possible lipid identifications?

This is a very interesting suggestion and we agree with the reviewer that future computational workflows should leverage as many information as possible for lipid identification, including lipid CCS values. Unfortunately, such a workflow is not available yet for timsTOF data. That said, we demonstrate the potential of such an approach with the example of lipid species that were not

annotated with our sample processing workflow (Figure 6 in our revised manuscript). This point also highlights the novelty of our lipidomics workflow, which leverages the speed of PASEF to, for the first time, fully characterize almost all detectable MS features by their retention time, accurate mass, collision cross section and fragment spectrum.

“A key feature of our workflow is that each detected MS feature is precisely positioned in the multi-dimensional data cuboid, while the speed of PASEF ensures that most of these features are associated with MS/MS information. We hypothesized that the combined information can be leveraged to assign putative lipid identifications for feature that would otherwise have remained unidentified. To test this, we overlaid all detected features with MS/MS information on top of all identified lipids (Fig. 6b). Zooming into the distinct space occupied by triglycerides revealed the conformational fine-structure of this lipid class, which results in clusters of lipids with the same acyl chain composition (Fig. 6c). Within each cluster, the lipids are differentiated by their degree of unsaturation as the addition of a double bond decreases the CCS value almost linearly. This enabled the identification of features that were not fully characterized by the available MS/MS information. As an example we putatively assign the MS feature at retention time 26 min, m/z 827.7115 ($\Delta = 1.9$ ppm) and CCS 311.2 Å² as TG 48:1 based on the relative position in the 3D space. This is further supported by the predicted CCS value of 308.4 Å², which deviates <1% from our experimental value. Similarly, we derived a putative assignment for TG 60:2, which had escaped identification due to a low-quality MS/MS spectrum in this particular experiment.”

7. Figure 2 caption: What elution time is shown for Figure 2a and 2b?

Thank you. We added this information to the figure legend.

8. Results, pg 8: In the 111,122 MS/MS spectra, how many of the peaks were analyzed multiple times? How can I use this number to evaluate the method?

In response to this point we now provide further details on the number of MS/MS spectra and selected precursors. To facilitate the evaluation of PASEF, we compare it to a conventional (non-PASEF) method in terms of speed and sensitivity:

“In this example, 16 precursors were selected during a single PASEF scan, which translates into a 16-fold increased MS/MS acquisition rate of over 100 Hz. Importantly, this does not come at a loss in sensitivity because the full precursor ion signal of the 100 ms accumulation time is captured.

In a 30 min analysis of plasma, we found that on average 15 precursors were fragmented per PASEF scan (Fig. 2c), confirming that the PASEF principle is transferable to lipidomics. In total, we acquired 187,177 MS/MS spectra - 15-fold more than without PASEF. This fragmentation capacity greatly exceeds the number of expected lipids and, in principle, allows to acquire MS/MS spectra for every suitable isotope pattern detected in a single lipidomics LC run. Here, we chose

to fragment low-abundance precursors repeatedly to increase their signal-to-noise ratios in a summed spectrum. On average, precursors were fragmented two times as indicated by the acquisition engine.”

9. Results, pg 8: What did acquiring each PASEF scan twice using two different collision energies accomplish? Couldn't the authors use different collision energies based on the drift time location of the ions like the energy ramping other IMS-MS/MS systems utilize?

The reviewers' comment led us to revisit our collision energy settings. In the course of the revision, we have refined our acquisition method and now use only one 'base' collision energy for each PASEF scan, while increasing the collision energy as a function of the ion mobility. The additional benefit of using two different base collision energies to generate a wider range of fragment ions was not substantial in our current data analysis workflow and we therefore did not use this strategy in our revised manuscript. We removed this sentence and updated the methods section accordingly.

“The ion mobility was scanned from 0.6 Vs/cm² to 1.95 Vs/cm². Precursors for data-dependent acquisition were isolated within ±1 Th and fragmented with an ion mobility-dependent collision energy, which was linearly increased from 25 eV to 45 eV in positive mode, and from 35 eV to 55 eV in negative mode.”

10. Results, pg 11: What confidence score method was utilized for the lipid identifications? How did the authors know when to “manually” remove a false positive?

Thank you. We have extended the description of our scoring method and added detail to our criteria for rejecting automatically annotated spectra.

“We manually inspected all automatically annotated MS/MS spectra to filter potential false positives based on the observed fragmentation pattern (Methods).”

“Lipid annotation of detected molecular features with assigned MS/MS spectra was performed using the ‘high-throughput lipid search (HTP)’ function of SimLipid v6.05 software (PREMIER Biosoft, Palo Alto, USA). The lipid search comprised four lipid categories, Glycerolipids (GL), Glycerophospholipids (GP), Sphingolipids (SP) and Sterol lipids (SL) and TAG, DAG, PA, PC, PE, PG, PI, PS, Ceramides, Sphingomyelins, Neutral Glycosphingolipids, Steryl esters, Cholesterols and Derivatives, as well as oxidized glycerophospholipids classes. PE and PC lipids with ether- and plasmalogen- substituents were considered. Lipid species from TAG and sterol classes were not considered for the negative mode MS/MS database search. Glycerophospholipids were only considered if containing an even number of carbons on at least one of the fatty acid chains. We searched for [M+H]⁺, [M+Na]⁺, and [M+NH₄]⁺ ions in positive mode, and [M-H]⁻, [M+Cl]⁻, [M-CH₃]⁻, [M+HCOO]⁻ and [M+AcO]⁻ in negative mode. The precursor ion and MS/MS fragment mass tolerances were set to 5 and 10 ppm, respectively.

The initial search results were filtered to ensure that lipids were annotated based on high-quality MS/MS spectra with fragment ions corresponding to structure specific characteristic ions. To this

end, we manually inspected the SimLipid results and removed potential false positives and refined lipid annotations based on head-groups and/or fatty acyl composition as follows. In positive mode, we rejected unlikely PC lipid ion species such as $[PC+NH_4]^+$ and $[PC+Na]^+$ if the corresponding $[M+H]^+$ ion was not observed, and if $[M-59+Na]^+$ (neutral loss of $(CH_3)_3N$) and $[M-183+Na]^+$ (neutral loss of phosphocholine) fragments were not detected. Lipids from GP, ST, and SP categories were required to have their corresponding head group diagnostic ions e.g., m/z 369.3516 for cholesterol esters, m/z 184.073 for PC lipid species, as well as the neutral loss of 141 Th for PEs. Neutral glycosphingolipids and ceramides were rejected if the structure-specific N^+ -type fragments were not annotated. However, lipid species from sterol classes were accepted if the precursor ion was the base peak in the MS/MS spectrum. TG/DG lipids with three or two unique fatty acid chains were reported only if at least two/one fatty acid chain fragment ion were/was detected. In negative mode, we rejected all lipid annotations for which we did not detect at least one characteristic fragment ion corresponding to one of the fatty acid chains.

We report lipid identifications with increasing level of fragment ion evidence using the following nomenclature: (i) a short name (e.g. PC 32:1) to indicate mass-resolved lipid molecular species, (ii) a long name for composition-resolved identifications where the symbol @ indicates that this particular acyl-chain is not fully characterized by fragment ions (e.g. Cer d18:1_26:0@), and (iii) a long name where head group and fatty acyl-chain composition are fully characterized (e.g. PG 16_1:16:1). Note that sn1/sn2/sn3 chain assignments, positions of the double bonds, as well as cis/trans isomers are not evident from our data and therefore not annotated.

11. Results, pg 13: I was confused by the comparison to the other studies. When the authors said they identified 76.2% and 71.2% of all glycerolipids and glycerophospholipids reported in both studies, how was that possible unless each study identified completely different lipids. From Figure 4, it looks like this study identified ~280 glycerophospholipids while Bowden identified ~125 and Quehenberger identified ~160. Even if they had no overlap in lipid identification that is only ~285 total lipids for comparison. Clarification of this whole paragraph is needed as I cannot believe there was no overlapping lipids in the two studies.

We agree with the reviewer that this paragraph in our initial manuscript was easily misunderstood and we have therefore rephrased this section. Please note that with our refined workflow, we now outperform both reference studies by large margins in all lipid categories except for ceramides (as discussed the manuscript) while using only a fraction of the starting material.

“Taking these two studies as a reference, we first compared the number of identified lipids in each lipid category based on the short name annotation (Fig. 4a and Suppl. Table S6-S7). Starting from 1 μ L plasma and with a single extraction protocol, our PASEF workflow detected many more glycerolipids and glycerophospholipids, exceeding both studies three- to four-fold. At the same time, 87% and 83% of all glycerolipids and glycerophospholipids reported in the Bowden study were also present in our dataset. Similarly, we retrieved 65% and 49% of all lipids from these two abundant plasma lipid categories reported by Quehenberger et al. We observed a two-fold gain

for sphingomyelins, again with a high overlap of 77% and 60% with both reference studies. Analysis of ceramides typically requires specific extraction methods and this category was therefore underrepresented in ours as well as in the Bowden study relative to the class-specific analysis by Quehenberger et al. From another analytically challenging class of lipids, sterol lipids, we still detected 33 species in the human plasma reference sample.”

12. Results, pg 14: If the authors inject twice as much sample into their instrument, how many lipids are they able to identify? Thus, what is the loss for having such a small injection?

We thank the reviewer for this interesting suggestion. In fact, injecting twice as much slightly decrease the number of identified lipids (up to 20%). This is because these are already relatively high sample amount on a nanoscale column, which can lead to peak broadening and shifts in the retention time, which in turn cause a decrease in identifications (see Figure just below). Instead, and to investigate the relation between sample load and lipid identifications in more detail, we performed a dilution series experiment (see also our response to point 5 above), which is now part of our revised manuscript.

Figure. Extracted ion chromatograms of five representative lipids (a-e) from human plasma with 1 μL (blue traces) and 2 μL (red traces) sample injection on the nanoflow column.

REVIEWERS' COMMENTS:

Reviewer #1 (Remarks to the Author):

I believe the authors have carefully considered the comments from the last round of review and the manuscript is significantly stronger. I have one request for revision prior to publication for the presentation of new data added to the manuscript. In Figure 5 panels d and e, the x-axis should be the same in both panels and the bin widths used for histogramming should also be the same. Presumably this will show a second comparison, namely that of the literature values and predicted values and the correspondence between the two. I would also recommend that the labels be expressly written in full form, rather than abbreviations (e.g. "(exp-liter)"..."(experimental-literature)").

With these changes I recommend publication in Nature Communications.

Reviewer #2 (Remarks to the Author):

The second version of this manuscript was greatly improved and much easier to understand. The authors addressed almost all of my comments and went beyond to fully validate CCS and RT CVs. I still believe there are two areas that need to be commented on by the authors and once addressed, their paper should be of interest to a wide audience of Nature Communications readers. The two areas are noted below.

1. In the rebuttal, the authors said that they profoundly disagreed with my assessment of little novelty versus the recently published proteomic papers using the same PASEF technique. Furthermore, they said the capability to fragment virtually all detected peaks and routinely reduce sample amounts by hundred-fold will have far reaching consequences in the future and that they had already seen this in their clinical work, where they can now measure lipidomic profiles for anything that was supplied for proteomics only.

My question back to them would be, what in this technique is different from the proteomic papers and specific to the lipidomic analyses shown? This is the main element of the manuscript that I am missing in order for it to be novel in my mind. Also, since lipidomic analyses often use much lower sample amounts than proteomics, wouldn't this be even more important for proteomic analyses (to

help reduce proteomic sample requirements)? These areas need to be addressed in the manuscript as there have been so many papers on PASEF lately, novelty in this manuscript needs to be shown.

2. In my original review, I noted that the authors never called out any novel isomers that were detected using the IMS. I would like to know if they found any or if they are just using IMS for filtering and an additional separation dimension in their evaluations. This should also be noted in the manuscript.

Point-by-point answers to the reviewer comments on “Trapped ion mobility spectrometry and PASEF enable in-depth lipidomics from minimal sample amounts”

We thank the referees very much for their time and careful revision of our manuscript. We have addressed the remaining points as detailed below and formatted the manuscript according to the editor’s instructions.

Reviewer #1 (Remarks to the Author):

I believe the authors have carefully considered the comments from the last round of review and the manuscript is significantly stronger. I have one request for revision prior to publication for the presentation of new data added to the manuscript. In Figure 5 panels d and e, the x-axis should be the same in both panels and the bin widths used for histogramming should also be the same. Presumably this will show a second comparison, namely that of the literature values and predicted values and the correspondence between the two. I would also recommend that the labels be expressly written in full form, rather than abbreviations (e.g. "(exp-liter)"..."(experimental-literature)").

With these changes I recommend publication in Nature Communications.

We are delighted that we were able to address all comments from the last round of review to the referee’s satisfaction. We thank the reviewer for the additional suggestions to clarify Figure 5. We made all changes as suggested.

Reviewer #2 (Remarks to the Author):

The second version of this manuscript was greatly improved and much easier to understand. The authors addressed almost all of my comments and went beyond to fully validate CCS and RT CVs. I still believe there are two areas that need to be commented on by the authors and once addressed, their paper should be of interest to a wide audience of Nature Communications readers. The two areas are noted below.

We highly appreciate the reviewer’s kind feedback and positive evaluation of our revised manuscript. We address the two remaining points directly below.

1. In the rebuttal, the authors said that they profoundly disagreed with my assessment of little novelty versus the recently published proteomic papers using the same PASEF technique. Furthermore, they said the capability to fragment virtually all detected peaks and routinely reduce sample amounts by hundred-fold will have far reaching consequences in the future and that they

had already see this in their clinical work, where they can now measure lipidomic profiles for anything that was supplied for proteomics only.

My question back to them would be, what in this technique is different from the proteomic papers and specific to the lipidomic analyses shown? This is the main element of the manuscript that I am missing in order for it to be novel in my mind. Also, since lipidomic analyses often use much lower sample amounts than proteomics, wouldn't this be even more important for proteomic analyses (to help reduce proteomic sample requirements)? These areas need to be addressed in the manuscript as there have been so many papers on PASEF lately, novelty in this manuscript needs to be shown.

The reviewer notes correctly that we reported the PASEF acquisition method and its advantages for proteomics in 2015 and 2018 as referenced in the introduction. Since then, our development has caused quite an impact in the proteomics community and is now making its way into many laboratories. However, to the best of our knowledge, those are currently the only two peer-reviewed publications in which PASEF has been applied. We anticipate that many more will follow as the technology spreads and we are convinced that the present manuscript marks a new milestone in this further development. For the first time, we are able to demonstrate the advantages of PASEF beyond proteomics, which we believe bears novelty on its own. Building on the unique advantages of PASEF, we developed a lipidomics workflow that greatly outperforms conventional approaches in terms of sensitivity and coverage. In addition to the technological advance, we believe our work also presents a conceptual advance in lipidomics research. This is because of the very unique situation in lipidomics in that the number of currently detectable lipids is in the range of hundreds to thousands. With the speed of PASEF we can now for the first time fragment virtually all detectable lipids in each sample in each single LC-MS run. Notably, this is very different from the situation in proteomics. Excitingly, in addition to this, we get the full benefits from the ion mobility separation and highly precise CCS information. All of this goes beyond conventional approaches and opens up many new research directions, both computationally and experimentally, and should therefore be of high interest to a broad audience, as acknowledged by the referee.

2. In my original review, I noted that the authors never called out any novel isomers that were detected using the IMS. I would like to know if they found any or if they are just using IMS for filtering and an additional separation dimension in their evaluations. This should also be noted in the manuscript.

The primary focus of the present manuscript is the PASEF acquisition method, which utilizes the TIMS device as an 'ion handling device' to sort precursors according to their ion mobility and compress the signal into narrow ion mobility peaks. As noted by the reviewer and highlighted in the introduction (see for example ref. 32), the TIMS separation itself can be further used to investigate isomeric species. However, in our opinion, a conclusive call of 'novel isomers' would

require much stronger and eventually targeted experimental evidence, which is beyond the scope of our untargeted workflow. For this reason, we state in the main text:

“Finally, we grouped adducts, isomers and co-eluting peaks that were separated by their ion mobility but could not be distinguished based on their MS/MS spectra.” (p. 11)

That said, we hope that our efforts to make all data publicly available encourages other researchers to mine them, for example, to investigate lipid isomers in more detail.